# DeL: Biologically Plausible Dendritic Learning Enables Class-Incremental Learning

## Abstract

Class-incremental learning (CIL) enables models to acquire new knowledge while retaining prior knowledge, thereby adapting to continuous data streams. Because parameter drift and distribution shifts are inevitable, it suffers from catastrophic forgetting and the stability–plasticity dilemma. There are various strategies to address these challenges. Nevertheless, they remain limited by the *homogeneous representations*, which reduce inter-class diversity and exacerbate forgetting. To overcome this bottleneck, we introduce dendritic learning (DeL), a biologically inspired framework that reduces homogeneous representations and thereby mitigates catastrophic forgetting. DeL leverages synaptic plasticity and multi-branch dendrites to extract diverse, discriminative features, fostering heterogeneous representation learning. A membrane layer integrates these features, and a subsequent somatic layer adapts them for downstream classification. By strengthening class-specific features, DeL also promotes robust memory consolidation. Experiments show that augmenting state-of-the-art CIL methods with DeL consistently boosts accuracy. Furthermore, DeL encourages more efficient representation learning, allowing the model to rely on fewer discriminative features. Code is available at https://github.com/anonymous/DeL.

## 1 Introduction

Artificial intelligence has achieved significant success across a wide range of real-world applications. Numerous specialized models have been developed to address specific tasks. However, a model that performs well on one class often struggles with another because of feature and parameter shifts (Zhou et al., 2024b). To maintain performance, such models typically require retraining or fine-tuning through transfer learning, which incurs additional computational cost. Real-world scenarios frequently produce non-stationary data streams that demand a model capable of continuously learning novel classes, called class-incremental learning (CIL). However, CIL faces the chronic problem of catastrophic forgetting, whereby the model gradually loses previously learned knowledge while assimilating new information. In contrast, humans exhibit a remarkable ability to preserve and integrate knowledge over time.

To achieve human-level learning, researchers have proposed various strategies to mitigate catastrophic forgetting. Because the loss of earlier knowledge is its principal cause (Robins, 1995), rehearsal-based CIL methods allocate a buffer that stores salient examples of past data, drawing inspiration from human memory consolidation (Zhuang et al., 2022). The model periodically revisits these samples while acquiring new information. Nevertheless, in conditions where historical data are scarce or privacy constraints preclude storage, rehearsal buffers cannot sufficiently preserve old knowledge, thereby degrading performance. Besides, parameter drift also contributes to forgetting. Expansion-based CIL approaches (Zhou et al., 2024a; Wu et al., 2022) therefore instantiate multiple lightweight subnetworks, each dedicated to a subset of classes, to minimize inter-class parameter drift. Although effective, both rehearsal- and expansion-based methods entail substantial computational overhead, the former is limited by buffer capacity, whereas the latter suffers from unbounded model growth. Parameter-regularization approaches (Kirkpatrick et al., 2017) first estimate how much each parameter contributes to previously learned tasks, then actively constrain the parameters they judge critical, thereby preserving prior knowledge. Although conceptually appealing, these methods perform poorly in CIL because importance estimates obtained at one incremental step often conflict with those required later (Van de Ven et al., 2022). Currently, knowledge-distillation

techniques rank among the most topical research directions for mitigating forgetting in CIL (Li et al., 2025). In this paradigm, a model trained on earlier tasks acts as a teacher that transfers its learned representations to a student model as the latter acquires new classes. CIL typically adopts a self-distillation variant. The teacher and student share an identical backbone while maintaining separate classification heads, enabling the teacher's predictions to guide the student and incorporate new knowledge without erasing the old. Although numerous strategies have been proposed to alleviate catastrophic forgetting by balancing the acquisition of new information with the retention of old knowledge, they still struggle to resolve the stability–plasticity dilemma, resulting in a persistent performance gap between artificial models and humans.

These limitations arise primarily from the biologically implausible architectures of conventional artificial neural networks (Bellitto et al., 2024). Contemporary models employ oversimplified Mc-Culloch–Pitts (MCP) neurons that interact solely through weighted summations. Consequently, their plasticity and stability depend almost entirely on architectural design and parameter values, especially in classifiers based on fully connected networks. This homogeneous processing drives successive layers to generate highly correlated activations, hindering any principled trade-off between stability and plasticity. In contrast, the human brain consolidates memories through precise synaptic and dendritic connectivity patterns, thereby forming stable long-term engrams (Ryan et al., 2015). Synaptic plasticity, which modulates the strength of these connections, provides the physiological substrate that enables humans to assimilate new information while preserving prior knowledge (Mansvelder et al., 2019). Motivated by this evidence, we argue that embedding biologically inspired mechanisms, specifically, synaptic plasticity for rapid adaptation and structured connectivity for durable memory, could markedly narrow the stability–plasticity gap in class-incremental learning.

To this end, we propose a biologically plausible dendritic learning (DeL) for CIL. DeL models neurons with a more realistic structure than the traditional MCP abstraction, incorporating a synaptic layer, a dendritic layer, a membrane layer, and a somatic layer. The synaptic layer receives features from the backbone and promotes diverse activations to highlight discriminative features. The dendritic layer is composed of multiple branches that repeatedly process synaptic signals with distinct plasticity profiles, thereby sustaining long-term potentiation, which is an essential mechanism for memory formation (Toni et al., 1999). Finally, the membrane and the somatic layers integrate the dendritic responses to produce class-specific outputs. DeL is model-agnostic and can be attached to any CIL method. We integrate DeL into several representative CIL paradigms and observe consistent accuracy improvements, confirming its ability to mitigate catastrophic forgetting. We further evaluate DeL on multiple benchmarks to characterize its stability–plasticity trade-off and conduct extensive ablation studies to quantify the contribution of each architectural component. Collectively, these results demonstrate that a biologically plausible dendritic learner can extract discriminative features from streaming data and perform robust incremental classification.

## 2  RELATED WORK

**Class-incremental Learning (CIL)** seeks to prevent catastrophic forgetting while balancing model stability and plasticity. Prior work can be summarized into four main categories. Rehearsal-based methods mimic human complementary learning by periodically revisiting a small buffer of past samples. The key challenge is selecting representative exemplars that preserve the discriminative structure of previous tasks. Recent work adopts multi-criteria selection to identify such exemplars (Rolnick et al., 2019; Zhuang et al., 2022). Parameter-regularization methods estimate the importance of each parameter to previously learned classes and constrain important ones during subsequent training. EWC (Kirkpatrick et al., 2017) introduces this paradigm, storing a Fisher-based importance matrix that matches the network's dimensionality. Expansion-based methods dynamically grow their architectures, allocating new capacity for incoming tasks to mitigate parameter drift. DER (Yan et al., 2021) appends an additional backbone for each new task and aggregates features through an enlarged fully connected layer. Although such methods often achieve state-of-the-art accuracy, they demand substantial memory and computation. Knowledge-distillation methods transfer information from a frozen "teacher" to a "student", learning new classes. LwF (Li & Hoiem, 2018) first applied logits-based distillation to align outputs of the model across tasks. iCaRL (Rebuffi et al., 2017) augments it with prototype rehearsal exemplars. Subsequent work incorporates feature-level distillation to preserve intermediate representations. TagFex (Zheng et al., 2025) combines

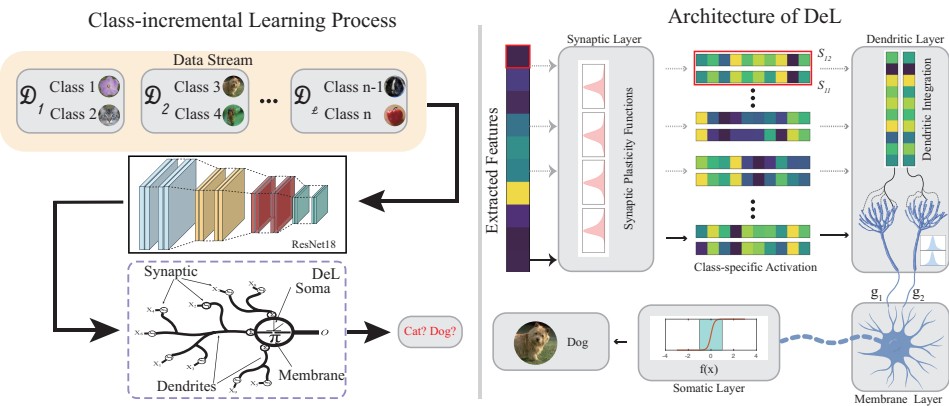

Figure 1: Overview of class-incremental learning with DeL. *Left*: The CIL process incorporating DeL. *Right*: The detailed architecture of the DeL module.

task-agnostic expansion with simultaneous feature and logits distillation to encourage diverse representation learning.

Collectively, these four research streams provide complementary perspectives on mitigating forgetting, yet each still faces trade-offs among accuracy, efficiency, and biological plausibility. By contrast, our method introduces a fully articulated dendritic architecture that leverages synaptic plasticity to boost feature diversity and safeguard long-term knowledge. Furthermore, its dendritic branches adaptively modulate the connection strengths between synapses and dendrites, enabling the model to extract discriminative features more effectively during incremental learning.

## 3 METHOD

### 3.1 PROBLEM DEFINITION

CIL seeks to build a unified classifier that can assimilate new classes from a continuously evolving data stream while preserving knowledge of previously learned classes. Let the data stream be $\mathbf{D} = \{\mathfrak{D}_1, \mathfrak{D}_2, \ldots, \mathfrak{D}_L\}$, where the $l$th incremental batch is $\mathfrak{D}_l = \{(\mathbf{x}_i^l, y_i^l)\}_{i=1}^{n_l}$, containing $n_l$ samples. Each instance $\mathbf{x}_i^l \in \mathbb{R}^D$ is paired with a label $y_i^l \in Y_l$, in which $Y_l$ is the label sets of incremental class $l$. Notably, $\forall i \neq l, Y_i \cap Y_l = \emptyset$. A corresponding test set is $\mathbf{D}^{\mathrm{t}} = \{\mathfrak{D}_1^t, \mathfrak{D}_2^t, \ldots, \mathfrak{D}_L^t\}$. Under the non-rehearsal assumption, samples from earlier increments are not reused when learning subsequent ones. After the $l$th increment, the model is evaluated on the cumulative label set $\mathcal{Y}_l = \bigcup_{k=1}^l Y_k$, thereby measuring its ability both to acquire new knowledge and to retain what it has previously learned.

### 3.2 DeL: BIOLOGICALLY PLAUSIBLE DENDRITIC LEARNING

Our method is grounded in neurophysiological evidence showing that behavioral learning leads to changes in synaptic strength, and that manipulating synaptic strength can alter the information stored in memory (Whitlock et al., 2006; Nabavi et al., 2014). Synaptic plasticity mechanisms that enhance synaptic strength contribute not only to the formation of memory traces but also to their reactivation during memory recall (Ryan et al., 2015; Nonaka et al., 2014). A detailed review of dendritic learning is described in supplementary file. We embed these mechanisms in a dendritic learning (DeL) network to emulate human learning and memory formation, as well as to confront the challenges of CIL. DeL comprises four functional layers, i.e., synaptic, dendritic, membrane, and somatic, working in concert to balance plasticity and stability throughout incremental learning, as shown in Figure 1. Moreover, conventional models process information homogeneously, therefore, they must learn many redundant features to separate classes, which strains memory capacity (Jaini et al., 2018). Once parameter or distribution shifts, such models struggle to generate reliable features, leading to rapid forgetting. Mitigating these shifts typically requires either adopting archi-

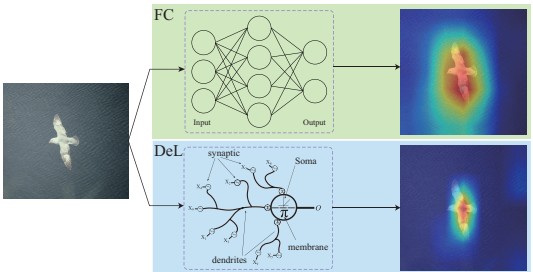

Figure 2: Grad-CAM (Selvaraju et al., 2017) visualizations for the fully-connection (FC) and DeL networks. The upper image corresponds to the FC model, and the lower image to DeL. DeL concentrates on highly discriminative regions, thereby reducing memory pressure.

tectures with additional parameters or using more distinctive features, both of which lessen memory pressure and diminish catastrophic forgetting. DeL's biologically inspired architecture and adaptive synaptic scaling extract discriminative features with far less redundancy. Figure 2 highlights this advantage. DeL is attentive to highly distinctive features than conventional ones, and utilizes them to make accurate classification. In addition, DeL is model-agnostic and can be seamlessly integrated into any CIL method by replacing the original logit classifier with DeL. In the case of knowledge distillation–based models, only logit distillation through DeL is required, without the need for any additional modifications.

### 3.2.1 SYNAPTIC LAYER

The layer ingests the feature stream produced by the backbone at every incremental step. To overcome the limitation of homogeneous representations, it must discriminate and refine task-specific features. Because class distributions differ, treating all inputs uniformly degrades performance. Biological neurons modulate synaptic strength in response to each input, amplifying salient signals and attenuating less informative ones, before forwarding them to the dendrites. We emulate this synapse plasticity by applying a learnable sigmoid function that maps each feature to the interval of $(0, 1)$. Meanwhile, considering the distribution difference of activations from backbone, we utilize the learnable function to evaluate their synaptic strength, i.e.,

$$S_{i,j} = \frac{1}{1 + e^{-(w_{i,j}x_i - \theta_{i,j})}} \tag{1}$$

where $w_{i,j}$ and $\theta_{i,j}$ denote the weight and threshold parameters of the learnable sigmoid that modulates the synaptic strength of dendrite $j$, thereby producing diverse connection patterns. The details of the synaptic layer are given in the supplementary file.

### 3.2.2 DENDRITIC LAYER

The dendritic layer receives heterogeneous synaptic signals and executes complementary functions. The synaptic layer forces discriminative features and can down-weight less informative connections, thereby reducing the feature set. However, over-pruning still risks degrading classification accuracy. To balance selectivity and coverage, we introduce a dendritic layer with multi-branch that mirrors the anatomical complexity of biological dendrites. Given the same input, each branch applies a distinct pattern of synaptic plasticity, enabling the network to capture diverse functional combinations and lessening its reliance on any single feature subset. Dendritic layer integrates synaptic outputs as $D_j = \sum_{i=1}^{N} S_{i,j}$, where $S_{i,j}$ is the $i$th synaptic signal on dendrite $j$. $N$ is the input size. This repeated aggregation strengthens plasticity events, promoting robust learning and memory consolidation.

### 3.2.3 MEMBRANE LAYER

The membrane layer aggregates the outputs of the dendritic branches. Because each dendrite performs a distinct function, the membrane applies a weighted fusion, i.e., $I = \sum_{j=1}^{M} g_j D_j$, where $g_j$

is a learnable, branch-specific gain. The adaptive gain further strengthens the model's capacity to learn novel information while preserving previously acquired knowledge.

### 3.2.4 SOMATIC LAYER

The somatic layer performs the final transformation, adapting the aggregated signal to the requirements of the downstream task, $O = f(I - \Delta)$, where $f(\cdot)$ is the activation function and $\Delta$ is a learnable threshold that adjusts to different tasks. Typically, the somatic layer applies sigmoid functions to the outputs of the membrane layer, producing the logits for each class individually. In classification problems, these logits reflect the model's confidence in each class, with larger values indicating higher certainty. The gradient computation and stability analysis are given in supplementary file.

## 4 EXPERIMENTS

### 4.1 EXPERIMENTAL SETUP

#### 4.1.1 DATASETS

Following the work (Sun et al., 2025a), we use six datasets to verify the performance, including CIFAR100 (Krizhevsky et al., 2009), CUB200 (Wah et al., 2011), ImageNet-A (Hendrycks et al., 2021b), ImageNet-R (Hendrycks et al., 2021a), VTAB (Zhai et al., 2019), and OmniBenchmark (Zhang et al., 2022). These datasets contain 100, 200, 200, 200, 50, and 300 classes, respectively, and we construct incremental streams of 10, 10, 20, 20, 10, and 30 new classes per step, starting from an initial zero-class state. Their details are summarized in the supplementary file.

#### 4.1.2 BASELINE METHODS

To evaluate DeL's performance, we integrated it into six representative CIL algorithms. Fine-tune serves as a plain baseline that employs no forgetting-mitigation strategy. Replay represents rehearsal-based methods, whereas DER (Yan et al., 2021) follows an expansion strategy. LwF (Li & Hoiem, 2018) and iCaRL (Rebuffi et al., 2017) both rely on knowledge distillation, and TagFex (Zheng et al., 2025) combines knowledge distillation with rehearsal and dynamic expansion. More details of all models are provided in the supplementary file.

#### 4.1.3 IMPLEMENTATION DETAILS

For a fair comparison, we keep the training and evaluation protocols identical across all methods. Every model is implemented in PyTorch using the PyCIL (Sun et al., 2023) and trained from scratch on an NVIDIA RTX 3090 GPU with ResNet18 backbone. Optimization is performed with SGD (initial learning rate $= 0.1$, weight decay $= 5 \times 10^{-4}$). The initial task is trained for 200 epochs. We use the random seed of $\{1993, 1994, 1995\}$ to guarantee the consistency of data partition.

#### 4.1.4 MEASUREMENT METRIC

We assess model performance with two complementary metrics. One of them is the top-1 accuracy after the final increment ($A_L$), another one is the average incremental accuracy $\bar{A}$. Let $A_l(l = 1, \ldots, L)$ denote the top-1 accuracy measured after the $l$th increment on the cumulative test set that includes every class learned thus far. The average incremental accuracy is calculated as $\bar{A} = \frac{1}{L} \sum_{l=1}^{L} A_l$.

### 4.2 PERFORMANCE RESULTS

For each model, we compute both $\bar{A}$ and $A_L$, then compare those with the DeL-enhanced variants. The experimental results are summarized in Table 1. DeL boosts replay methods by 4.1% in $\bar{A}$ and 3.7% in $A_L$, and raises LwF, a knowledge-distillation approach, by 4.2% and 3.0%, respectively. It achieves improvements of 3.9% and 3.1% in iCaRL. As baselines, the expansion-based methods, DER and TagFex, outperform other types of methods, which indicates that larger parameters efficiently inhibit parameter shift and store more features to prevent forgetting. DeL further

Table 1: Average and last performance comparison on six datasets with **ResNet-18** as the backbone. We report all compared methods with their source-code provided by PyCIL library and trained from scratch. The best performance is shown in bold.

| Methods | CIFAR100 | | CUB200 | | VTAB | | ImageNet-A | | ImageNet-R | | OmniBench | |
|---|---|---|---|---|---|---|---|---|---|---|---|---|
| | $\bar{A}$ | $A_L$ | $\bar{A}$ | $A_L$ | $\bar{A}$ | $A_L$ | $\bar{A}$ | $A_L$ | $\bar{A}$ | $A_L$ | $\bar{A}$ | $A_L$ |
| Finetune | $25.94_{\pm0.14}$ | $9.28_{\pm0.22}$ | $20.28_{\pm1.02}$ | $10.53_{\pm0.39}$ | $41.91_{\pm2.82}$ | $26.82_{\pm2.20}$ | $6.74_{\pm0.70}$ | $2.50_{\pm0.14}$ | $15.18_{\pm0.85}$ | $7.05_{\pm0.48}$ | $24.41_{\pm0.06}$ | $8.70_{\pm0.14}$ |
| w/ DeL | $\mathbf{26.88}_{\pm0.11}$ | $\mathbf{9.55}_{\pm0.25}$ | $\mathbf{24.73}_{\pm0.90}$ | $\mathbf{12.15}_{\pm0.66}$ | $\mathbf{43.64}_{\pm1.79}$ | $\mathbf{27.83}_{\pm1.46}$ | $\mathbf{7.54}_{\pm0.47}$ | $\mathbf{2.79}_{\pm0.08}$ | $\mathbf{17.14}_{\pm1.55}$ | $\mathbf{8.01}_{\pm0.50}$ | $\mathbf{24.76}_{\pm0.13}$ | $\mathbf{8.83}_{\pm0.10}$ |
| Replay | $60.24_{\pm0.38}$ | $42.51_{\pm1.21}$ | $34.14_{\pm3.18}$ | $27.52_{\pm1.72}$ | $56.54_{\pm2.01}$ | $46.76_{\pm1.76}$ | $6.43_{\pm0.26}$ | $2.50_{\pm0.14}$ | $22.83_{\pm0.30}$ | $15.76_{\pm0.07}$ | $51.80_{\pm0.22}$ | $33.18_{\pm0.81}$ |
| w/ DeL | $\mathbf{63.91}_{\pm0.87}$ | $\mathbf{46.58}_{\pm0.82}$ | $\mathbf{42.94}_{\pm3.10}$ | $\mathbf{36.75}_{\pm2.12}$ | $\mathbf{60.53}_{\pm1.38}$ | $\mathbf{48.21}_{\pm2.38}$ | $\mathbf{7.23}_{\pm0.34}$ | $\mathbf{2.52}_{\pm0.36}$ | $\mathbf{29.25}_{\pm0.86}$ | $\mathbf{22.06}_{\pm0.77}$ | $\mathbf{54.73}_{\pm0.71}$ | $\mathbf{36.28}_{\pm0.54}$ |
| LwF | $46.56_{\pm0.56}$ | $25.55_{\pm0.54}$ | $24.34_{\pm2.24}$ | $12.85_{\pm0.72}$ | $41.80_{\pm4.10}$ | $25.03_{\pm1.54}$ | $7.50_{\pm0.76}$ | $3.45_{\pm0.30}$ | $20.24_{\pm1.18}$ | $9.91_{\pm0.76}$ | $38.40_{\pm0.55}$ | $18.26_{\pm1.10}$ |
| w/ DeL | $\mathbf{52.98}_{\pm0.79}$ | $\mathbf{31.08}_{\pm0.59}$ | $\mathbf{31.53}_{\pm1.47}$ | $\mathbf{18.43}_{\pm1.03}$ | $\mathbf{45.30}_{\pm4.51}$ | $\mathbf{27.63}_{\pm2.84}$ | $\mathbf{8.59}_{\pm0.22}$ | $\mathbf{3.73}_{\pm0.36}$ | $\mathbf{25.71}_{\pm1.73}$ | $\mathbf{13.71}_{\pm1.05}$ | $\mathbf{41.94}_{\pm0.73}$ | $\mathbf{19.84}_{\pm0.84}$ |
| iCaRL | $60.21_{\pm0.59}$ | $42.00_{\pm1.49}$ | $33.17_{\pm2.56}$ | $24.92_{\pm0.77}$ | $61.59_{\pm2.89}$ | $51.91_{\pm1.32}$ | $6.63_{\pm0.18}$ | $2.22_{\pm0.41}$ | $23.71_{\pm1.13}$ | $15.62_{\pm0.83}$ | $52.07_{\pm0.33}$ | $31.87_{\pm0.27}$ |
| w/ DeL | $\mathbf{63.55}_{\pm1.03}$ | $\mathbf{44.98}_{\pm1.44}$ | $\mathbf{40.65}_{\pm3.70}$ | $\mathbf{30.69}_{\pm2.88}$ | $\mathbf{64.20}_{\pm2.01}$ | $\mathbf{52.50}_{\pm2.47}$ | $\mathbf{7.08}_{\pm0.27}$ | $\mathbf{2.61}_{\pm0.11}$ | $\mathbf{28.56}_{\pm0.72}$ | $\mathbf{19.30}_{\pm0.66}$ | $\mathbf{53.94}_{\pm0.40}$ | $\mathbf{33.59}_{\pm0.36}$ |
| DER | $71.99_{\pm2.15}$ | $61.86_{\pm0.89}$ | $41.45_{\pm3.02}$ | $38.98_{\pm1.43}$ | $61.59_{\pm3.09}$ | $48.60_{\pm5.04}$ | $6.61_{\pm0.41}$ | $2.48_{\pm0.03}$ | $33.73_{\pm0.48}$ | $30.98_{\pm0.39}$ | $66.46_{\pm0.51}$ | $55.97_{\pm0.13}$ |
| w/ DeL | $\mathbf{73.92}_{\pm1.03}$ | $\mathbf{63.47}_{\pm0.35}$ | $\mathbf{47.55}_{\pm2.89}$ | $\mathbf{43.02}_{\pm1.21}$ | $\mathbf{65.97}_{\pm2.78}$ | $\mathbf{53.10}_{\pm2.29}$ | $\mathbf{7.27}_{\pm0.15}$ | $\mathbf{2.92}_{\pm0.17}$ | $\mathbf{39.34}_{\pm1.83}$ | $\mathbf{35.06}_{\pm1.71}$ | $\mathbf{67.70}_{\pm0.77}$ | $\mathbf{57.63}_{\pm0.62}$ |
| TagFex | $72.55_{\pm1.04}$ | $62.11_{\pm0.33}$ | $46.04_{\pm3.73}$ | $41.40_{\pm2.03}$ | $63.77_{\pm1.82}$ | $49.62_{\pm3.14}$ | $5.80_{\pm0.73}$ | $1.18_{\pm1.35}$ | $37.87_{\pm2.69}$ | $35.04_{\pm2.21}$ | $66.54_{\pm0.88}$ | $56.52_{\pm0.26}$ |
| w/ DeL | $\mathbf{74.96}_{\pm1.04}$ | $\mathbf{65.68}_{\pm0.01}$ | $\mathbf{49.05}_{\pm4.59}$ | $\mathbf{45.00}_{\pm2.50}$ | $\mathbf{65.82}_{\pm1.65}$ | $\mathbf{54.44}_{\pm2.03}$ | $\mathbf{6.78}_{\pm1.03}$ | $\mathbf{1.86}_{\pm1.07}$ | $\mathbf{43.96}_{\pm0.97}$ | $\mathbf{39.65}_{\pm1.40}$ | $\mathbf{67.92}_{\pm1.32}$ | $\mathbf{58.43}_{\pm0.82}$ |

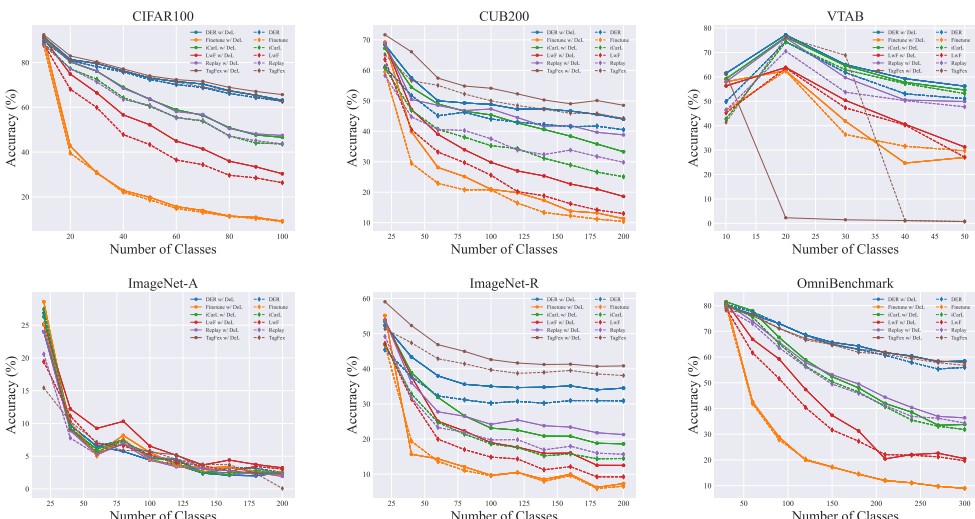

Figure 3: Performance curve of different methods with and without DeL. The solid and dot-line denote the variant with and without DeL, respectively.

Table 2: The ablation study of Synaptic Plasticity on CIFAR 100 with **DER**. The best performance is shown in bold.

| SA | SN | CIFAR100 | | VTAB | |
|---|---|---|---|---|---|
| | | $\bar{A}$ | $A_L$ | $\bar{A}$ | $A_L$ |
| ✗ | ✗ | 71.93 | 61.18 | 58.03 | 49.30 |
| ✗ | ✓ | 33.90 | 1.00 | 54.29 | 44.10 |
| ✓ | ✗ | **74.68** | **64.31** | **64.51** | **56.22** |
| ✓ | ✓ | 73.69 | 63.58 | 63.85 | 56.19 |

improves their performance. The experimental results strongly demonstrate that synaptic plasticity in DeL produces heterogeneous and highly discriminative representations, enabling the network to rely on a smaller yet more informative feature set and thus reducing memory overhead while mitigating catastrophic forgetting. Furthermore, we give the performance curve in Figure 3, showing that methods with DeL achieve a large performance gap compared to methods without it. Moreover, the long-sequence tasks, additional computation and train time are reported in supplementary file.

## 4.3 ABLATION STUDY

We conduct an ablation study on CIFAR-100 and VTAB to quantify the contribution of synaptic plasticity in DeL. The results are summarized in Table 2. **SA** denotes the learnable synaptic-

Table 3: The parameter analysis of DeL on VTAB.

| $M$ | DER | | iCarL | |
|---|---|---|---|---|
| | $\bar{A}$ | $A_L$ | $\bar{A}$ | $A_L$ |
| 1 | 63.73 | 54.91 | 64.67 | 50.20 |
| 2 | **64.51** | **56.22** | 67.02 | 49.03 |
| 3 | 64.43 | 56.09 | 63.58 | 47.69 |
| 4 | 62.54 | 55.80 | 64.93 | 49.04 |

Table 4: The backbone analysis on VTAB with **DER**. The best performance is shown in bold.

| Backbone | ResNet18 | | ResNet34 | | Pre-trained ViT | |
|---|---|---|---|---|---|---|
| | $\bar{A}$ | $A_L$ | $\bar{A}$ | $A_L$ | $\bar{A}$ | $A_L$ |
| DER | 58.03 | 51.07 | 55.26 | 47.79 | 91.74 | 92.05 |
| w/ DeL | **63.85** | **56.19** | **62.36** | **51.32** | **92.10** | **92.09** |

plasticity function, whereas **SN** performs input normalization through layer normalization (Ba et al., 2016). The results reveal that incorporating synaptic plasticity markedly improves performance of the model. Moreover, the modest drop when incorporating normalization is removed suggests that the synaptic layer accommodates substantial distribution shifts, allowing the model to learn effectively from streaming data. DeL therefore maintains strong performance even without explicit normalization. The detailed results of ablation study are given in supplementary file.

Dendritic morphology endows the network with greater computational capacity and robust learning dynamics (Chavlis & Poirazi, 2025). It also modulates plasticity; therefore, we examine how the number of dendritic branches $M$ affects performance, as shown in Table 3. In principle, increasing $M$ enhances synaptic plasticity and nonlinear expressiveness, yet excessive plasticity undermines stability. Our results show that $M = 2$ offers the best trade-off, whereas larger or smaller values lead to performance declines. Besides, we discuss the impact of DeL for different backbones (Sun et al., 2025b) in Table 4, indicating DeL possesses the advantages of model-agnostic, stability, and robustness. Moreover, pretrained-model–based approaches is discussed in supplementary file.

### 4.4 VISUALIZATIONS

We employ Grad-CAM (Selvaraju et al., 2017) to visualize model's attention regions and assess the impact of homogeneous representations. As Figure 4 illustrates, baseline iCaRL attends disproportionately to non-target regions, generating spurious memories that accelerate subsequent forgetting, which evidences that networks built on MCP neurons struggle to isolate class-relevant features due to homogeneous representation. In contrast, iCaRL enhanced with DeL consistently focuses on truly discriminative cues, most notably the bird's body and head, thereby preserving salient attention patterns and maintaining robust feature representations. This advantage is particularly pronounced at step $T_2$. These observations suggest that DeL effectively curbs representation drift and efficiently store memory, yielding a more favorable plasticity–stability trade-off.

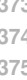
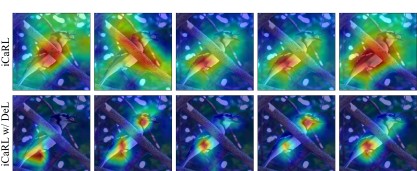

Figure 4: Grad-CAM visualizations across incremental tasks $T_1$ to $T_5$. *Top*: Grad-CAM results from iCaRL. *Bottom*: Grad-CAM results from iCaRL with DeL.

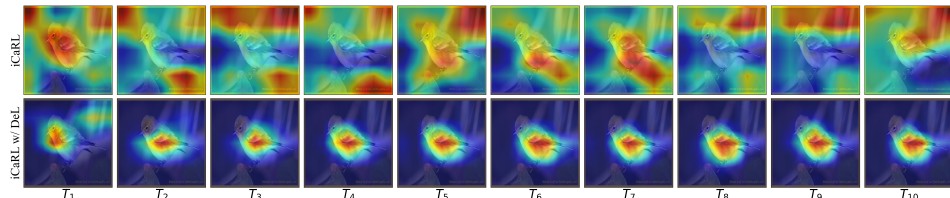

Figure 5: The Grad-CAM of activation maximization maps, which are prone to catastrophic forgetting due to their dependence on the classifier.

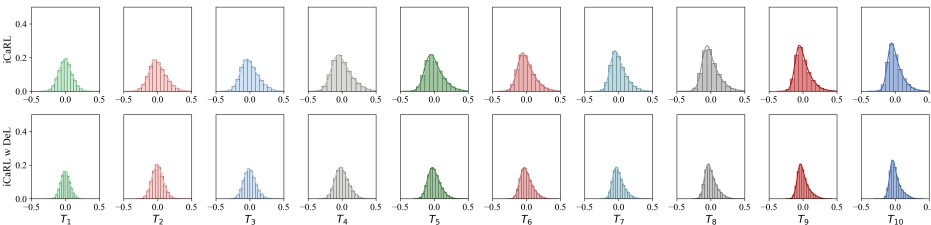

Figure 6: The weight probability density functions after class-incremental learning for iCaRL with and without DeL.

Furthermore, Figure 5 shows that the baseline iCaRL model correctly attends to the bird's body in the early tasks, however, as class-incremental learning progresses its attention drifts toward background regions, indicating catastrophic forgetting. By the final tasks, Grad-CAM no longer highlights critical features such as the head, chest, and wings, and the representations of earlier classes have markedly decayed. In contrast, once we embed DeL, the model's attention remains stable across all ten tasks. High-response regions (in red) consistently cover key features, chiefly the chest and back. Therefore, even after the final task the model localizes these regions while relying on fewer overall attention areas. This behavior evidences that DeL enhances neural selectivity, preserves memory, and effectively resists catastrophic forgetting by suppressing homogeneous representations and promoting truly discriminative features.

## 4.5 ANALYSIS OF EFFECTIVENESS

To better understand how DeL mitigates catastrophic forgetting, we analyze its weight distribution after each incremental step on the CUB dataset in Figure 6. From this figure, the weight distribution of conventional iCaRL shifts at every stage, revealing parameter drift that drives forgetting. By contrast, incorporating DeL can suppress the distribution drift to a certain extent. Importantly, DeL exhibits a more concentrated distribution, showing that DeL can efficiently utilize weights to constrain drift and concentrate model capacity on the most salient features via synaptic plasticity. Thereby it can preserve earlier knowledge while assimilating new classes.

Furthermore, we compute information entropy for each neuron to quantify its class specificity and importance for classification results. High entropy indicates mixed selectivity, meaning a neuron responds to multiple classes, whereas low entropy reflects class-specific tuning. Figure 7 plots the entropy distribution after each incremental step. The baseline model shows uniformly high entropy, suggesting that nearly all neurons participate in every classification decision and therefore struggle to preserve the class-specific features needed to recall old knowledge. In contrast, the DeL-augmented model displays a broader and often bimodal entropy distribution. Many neurons shift toward lower entropy values, demonstrating distinct activations for different classes, while a smaller subset retains higher entropy, preserving flexibility for new tasks. This adaptive redistribution implies that DeL dynamically adjusts the proportion of class-specific neurons as new classes arrive, which mitigates catastrophic forgetting. Specifically, the entropy distribution becomes bimodal at certain increments, suggesting that the model adjusts the proportion of class-specific neurons to match the current class composition. Detailed calculation procedures appear in the supplementary file. Overall, the results confirm that DeL's synaptic-plasticity mechanism fosters a richer pool of

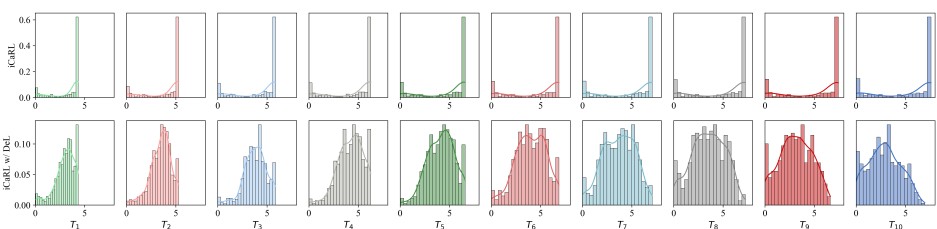

Figure 7: Information entropy distributions for iCaRL with and without DeL. Entropies are computed from classifier activations on all test images of the CUB dataset during class-incremental learning.

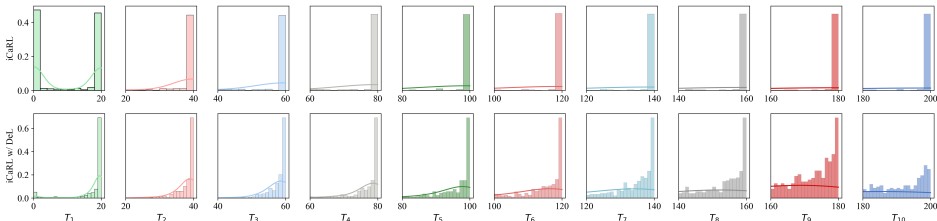

Figure 8: Probability density functions of class selectivity. Each histogram illustrates the distribution of neuron selectivity across classes at each incremental learning stage. The number of bins corresponds to the total number of classes in the CUB dataset.

class-specific neurons, enabling the network to remember discriminative features more effectively during incremental learning.

Besides, the entropy analysis shows that DeL yields a larger pool of class-specific neurons through synaptic plasticity, yet the exact specificity of these neurons remains uncertain. To clarify how many classes each neuron represents, we compute a selection index and apply a 0.5 activation threshold due to the use of sigmoid function. Figure 8 presents the resulting class-activation counts. At every incremental step, neurons in the baseline model activate predominantly for the most recently learned classes, with few responses to earlier classes. In contrast, neurons in the DeL-enhanced model respond to a broader range of classes; although the network still favors new information, it preserves considerably more activations for previously learned classes. These findings provide additional evidence that DeL effectively mitigates catastrophic forgetting.

## 5 CONCLUSION

We propose a biologically inspired dendritic learning module (DeL) designed to enhance class-incremental learning by alleviating the issues of catastrophic forgetting and homogeneous representation. DeL integrates synaptic plasticity and dendritic computation into a four-layer architecture that promotes diverse and discriminative feature extraction. Extensive experiments demonstrate that DeL improves the performance of representative CIL methods across multiple benchmarks. As model-agnostic, we discuss the performance of DeL by incorporating it into various types of CIL methods. Grad-CAM visualizations show that DeL captures discriminative features and directs more attention to task-relevant object regions. Because it requires fewer stored exemplars, DeL alleviates catastrophic forgetting. Furthermore, through ablation and analysis of effectiveness, we show that DeL reduces parameter drift, increases the proportion of class-specific neurons, and stabilizes attention to semantically meaningful image regions. These results confirm that embedding biologically plausible mechanisms into artificial neural networks can significantly enhance continual learning performance by improving feature selectivity and memory retention. Although DeL is advances CIL through biological mechanism, its present architecture lacks flexibility because its hyperparameters are fixed. We will explore dynamic DeL structures to enhance scalability in future work.

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
