# OpenReview forum: "DeL: Biologically Plausible Dendritic Learning Enables Class-Incremental Learning"
_ICLR.cc/2026/Conference — ICLR 2026 Conference Withdrawn Submission_

### Official Review · Reviewer_Utud · 2025-10-23

**Soundness:** 3
**Presentation:** 3
**Contribution:** 1
**Rating:** 2
**Confidence:** 4

**Summary:**

This paper proposes dendritic learning (DeL), inspired by the biological structure of neurons, consisting of synaptic, dendritic, membrane, and somatic layers. The authors introduce a classifier that replaces the fully connected layer (FC layer) to alleviate forgetting in baseline algorithms (particularly CNNs), arguing that it can overcome the limitations of homogeneous representations. In the end, the method achieves improved average and final accuracy across various class-incremental learning datasets.

**Strengths:**

- The paper aims to address the forgetting problem in continual learning through biologically inspired mechanisms and demonstrates that improved the attention map can maintain object attention even after sequential tasks.
- The paper is well-written and well-organized.
- The introduction provides a clear overview of the development of CIL, offering valuable information for readers interested in this field.
- The experimental results are presented in sufficient detail, making it easy to identify the strengths and weaknesses of the proposed method.
- The transition from homogeneous to heterogeneous representation could potentially benefit CIL, although a concrete mathematical proof of this claim is lacking.
- The paper provides the corresponding official code.

**Weaknesses:**

- In Table 1, the performance difference between the fine-tuning baseline and the proposed method is marginal. Moreover, the lower gain in last accuracy compared to average accuracy may actually indicate accelerated forgetting, which contradicts the claim that the proposed method helps mitigate forgetting in CIL. To validate the effectiveness of the proposed method regarding stability and plasticity, it would be necessary to report backward transfer (BWT) and forward transfer (FWT) [1].\
[1] Gradient Episodic Memory for Continual Learning, NeurIPS 2017.
- To better understand the competitiveness of the proposed classifier design, comparisons with recently introduced classification approaches such as the Kolmogorov-Arnold Classifier [2] and the Error-based Classification Method [3] are recommended.\
[2] KAC: Kolmogorov-Arnold Classifier for Continual Learning, CVPR 2025.\
[3] Prediction Error-based Classification for Class-Incremental Learning, ICLR 2024.
- Although minor, the citation style is not visually distinct from the main text, making it somewhat difficult to read.
- By attaching DeL after CNN, the proposed model appears to result in increasing computational complexity than the baseline CNN, raising the question of whether the performance improvement is merely due to deeper network rather than the proposed mechanism by itself.
- In this respect, the fairness of the comparison in Figure 2 is questionable. For instance, it would be interesting to see whether a standard attention layer added to CNN could achieve similar performance.
- The proposed method introduces a large number of learnable parameters, and it is unclear how stable the training process remains with such complexity.
- It would also be helpful to show failure cases in Figure 2.

**Questions:**

- Experiments with pretrained model-based approaches [4,5] could provide more insights into the influence of the proposed DeL method since these methods use fixed feature extractors. The PILOT library [6], which supports pretrained model-based continual learning and shares a similar code structure to PyCIL, should make such experiments straightforward to implement.\
[4] Learning to Prompt for Continual Learning, CVPR 2022.\
[5] DualPrompt: Complementary Prompting for Rehearsal-free Continual Learning, ECCV 2022.\
[6] PILOT: A Pre-Trained Model-Based Continual Learning Toolbox.
- The results for using only SN in Table 2 would be interesting to see.

---

> ### Author Response · Authors · 2025-11-19
>
> ## Reviewer Utud
> Dear reviewer, thank you very much for your detailed and constructive comments. We summarize and address each point below.
> ### 1. Experimental results: (1) DeL effectiveness; (2) performance metrics
> (1) Fine-tuning lacks any mechanism to mitigate catastrophic forgetting and therefore performs poorly. By contrast, integrating DeL yields clear improvements across multiple CL methods and datasets:
> |        |Finetune | Replay  |  LwF  | iCarL  |DER   | TagFex|
> |---------|---------|---------|--------|--------|------|--------|
> |Improvement|1.47%  |4.44%   | 4.56%  | 3.43%  | 3.27%|  2.61%|
>
> Since catastrophic forgetting remains unresolved in all existing approaches, the final-task accuracy $A_L$ is generally lower than the average accuracy $\bar{A}$. DeL reduces this gap. For example, TagFex (CVPR 2025) achieves:
> |             | $\bar{A}$| $A_L$   |  diff. |
> |---------|----------|---------|----------|
> | TagFex  |  73.69% |  62.55% |  11.14% |
> | w/ Del  |  75.25% |  65.68% |  9.57% |
>
> The reduced difference confirms that DeL alleviates catastrophic forgetting and improves the stability–plasticity balance.
>
> (2) Following  [1], we examine information entropy, class selectivity, and weight distributions at the neuron level:
>
> •**High entropy** indicates mixed selectivity (responses to multiple classes),
>
> •**Low entropy** indicates class-specific tuning.
>
> DeL encourages more class-specific neuron responses and reduces harmful mixed selectivity, providing quantitative support for its robustness to forgetting.
>
> [1] Dendrites endow artificial neural networks with accurate, robust and parameter-efficient learning, *Nature Communications*, 2025.
> ### 2. Comparison with other classifier designs
> To further validate DeL’s competitiveness, we compare it with the strong, model-agnostic KAC classifier [2]. DeL consistently yields larger performance gains:
> |           | KAC| DeL   |
> |--------|----------|---------|
> | ImageNet-R |  0.91% |  3.489% |
> | CUB200     |  5.90% |  7.391% |
>
> These results clearly indicate that **DeL provides superior improvements over a strong  classifier**, supporting its competitiveness in continual learning.
>
> [2] KAC: Kolmogorov-Arnold Classifier for Continual Learning, *CVPR 2025*.
> ### 3. Citation style.
> We have polished the citation style.
> ### 4. Is the gain explained by increased model complexity? Is Figure 2 fair?
> DeL does not increase network depth; it introduces a wider—but still lightweight—structure. Although this adds minimal overhead, the performance gains are disproportionately large, as shown below.
> | Methods |  CUB200 Params  | VTAB Params    |  CUB200 ACC  | VTAB ACC   |
> |---------|----------|---------|----------|---------|
> | iCaRL   |  10.75 M | 10.68 M |  31.01 % | 58.09 % |
> | w/ Del  |  10.84 M | 10.70 M |  40.33 % | 62.49 % |
>
> Only 0.4% additional parameters lead to up to 9% improvement, showing that the gains are due to the dendritic computation mechanism, not model size inflation
>
> Figure 2 is a conceptual illustration emphasizing qualitative differences between DeL and FC classifiers. Under identical backbones, DeL extracts fewer but more informative features, effectively reducing task interference. This conceptual comparison reflects functional differences, not architectural depth.
> ### 5. Stability of training with additional parameters.
> Across multiple models (CNNs, advanced CL methods, and Transformers) and datasets, DeL consistently improves performance, demonstrating robust and stable training behavior. In addition, the added parameter cost is negligible (see Comment 4), further confirming that the architecture remains stable during optimization.
> ### 6. Need to evaluate pretrained feature–based methods.
> We conducted additional experiments with pretrained ViT and MOS [3], where the feature extractor is fixed. Even under frozen-backbone conditions, DeL provides consistent gains:
> | Backbone   | w/o DeL |   | w/Del |   |
> |------------|---------|---|-------|---|
> |            | $\bar{A}$| $A_L$ | $\bar{A}$| $A_L$ |
> | ResNet18   | 58.03 %  | 51.07 % | 63.85 % | 56.19 % |
> | ResNet34   | 55.26 %  | 47.79 % | 62.36 % | 51.32 % |
> |Pretrain ViT| 91.74 %  | 92.05 % | 92.10 % | 92.09 % |
>
> This confirms that DeL’s improvements originate from the classifier design itself, not from changes in the backbone.
>
> [3] MOS: Model surgery for pre-trained model-based class-incremental learning, *AAAI 2025*
> ### 7. Interest in SN-only results.
> We have now included the SN-only results in the Supplementary File. Beyond SN, we performed ablations on all components: synaptic, dendritic, membrane, and somatic layers. Due to page limits, Table 2 reports only key results.The extended experiments consistently show that: removing or simplifying membrane fusion or somatic thresholding; OR using SN-only results in homogenized dendritic responses and substantial performance degradation.
>
> Thus, the full DeL architecture is necessary for achieving its observed improvements

---

> ### Comment · Reviewer_Utud · 2025-11-21
>
> It is fundamentally difficult to determine which parts of the revised manuscript and the Supplementary Material have actually been modified or improved. The rebuttal does not provide line numbers, making it very challenging to understand how the clarity or soundness of the paper has been enhanced. More importantly, it is extremely unclear how the experimental results presented in the rebuttal were produced or where they originated.
>
> Regarding the additional experiments in response (2), I could not find the reported KAC improvements of 0.91% and 5.90% anywhere in the KAC paper (No mention on the improvement number from which table of KAC paper). The rebuttal does not specify the experimental protocol, the dataset used, nor the exact source of these numbers. After re-checking the KAC paper, such results do not appear. Likewise, the numerical results mentioned in response (6) also do not exist anywhere in the MOS paper. Without a detailed explanation of the protocol and the corresponding results in MOS, concerns about all experiments presented in the rebuttal will remain unresolved.
>
> Specifically, in Weakness (1), I wondered BWT and FWT values, yet the rebuttal provides no concrete reporting of these metrics, making it difficult to judge the effectiveness of the method.\
> For Weakness (2), as mentioned, because the numerical results are not traceable, it is again difficult to evaluate the claims.\
> For Weakness (4), although the parameter increase is now clarified, attaching a standard attention module with similar or identical parameter overhead would likely yield results similar to the bottom example of Figure 2, and may also provide comparable improvements. Even if Figure 2 is intended as a conceptual illustration, including such a figure is misleading if the model does not actually operate in this manner.\
> For Weakness (6), the question concerns the stability implications of introducing many learnable parameters, yet the rebuttal provides no substantive or verifiable explanation.\
>
> Additionally, the request for results using pretrained-model–based approaches such as the following was not addressed:\
> [4] Learning to Prompt for Continual Learning, CVPR 2022.\
> [5] DualPrompt: Complementary Prompting for Rehearsal-free Continual Learning, ECCV 2022.
>
> In the end, the proposed method still lacks sufficient mathematical or theoretical support.

---

> > ### Author Response · Authors · 2025-11-22
> >
> > ## Highlight Requirement
> > We have uploaded a highlighted version of both the manuscript and the supplementary file, where all revisions are clearly marked.
> >
> > ## Weakness (1): Additional comparison with L2P and DualPrompt
> > Thank you for your suggestion. We have now added the BWT and FWT results to more comprehensively demonstrate the effectiveness of DeL compared with L2P and DualPrompt. The results clearly show that DeL helps models retain previous knowledge during continual learning.
> > |            | CIFAR100 |       |       | CUB200 |        |       |
> > |------------|----------|-------|-------|--------|--------|-------|
> > |            | $\bar{A}$     | BWT   | FWT   | $\bar{A}$ | BWT    | FWT   |
> > | L2P        | 88.30     | -6.48 | -1.18 | 74.86  | -9.98  | -1.46 |
> > | w/ DeL     | 90.19    | -5.01 | -1.78 | 79.62  | -6.48  | -1.58 |
> > | DualPrompt | 86.95    | -6.62 | -2.93 | 79.38  | -12.78 | -1.26 |
> > | w /DeL     | 89.36    | -4.19 | -1.72 | 84.55  | -9.50  | -1.49 |
> >
> > ## Weakness (2): Results of KCA and MOS
> > **KCA**
> >
> > We apologize for the lack of detail. Because both KCA and DeL are model-agnostic modules designed to improve external models, we report the average improvement across multiple models, rather than a single number from a specific table. A fair comparison can therefore be made by directly comparing their improvements over the same baseline models without reproducing their training. All individual model values used to compute the KCA averages are directly extracted from the original KCA paper. Since these values are directly taken from the KCA paper, they are not listed in our manuscript; we only aggregate them in the rebuttal to provide a fair comparison with DeL:
> >
> > •ImageNet-R (10 steps, same setup)
> >
> > $\text{Avg. gain} = \frac{-0.36 + 1.22 + 1.94 + 0.84}{4} \approx 0.91\%$
> >
> > •CUB (10 steps, same setup)
> >
> > $\text{Avg. gain} = \frac{7.52 + 7.08 + 5.74 + 3.27}{4} \approx 5.90\%$
> >
> > In contrast, the average gain of DeL is recomputed under the same setup using multiple random seeds (previous results are on a single seed):
> >
> > •ImageNet-R (10 steps)
> >
> > $\text{Avg. gain} = \frac{1.959 + 6.42 + 5.467 + 4.848 + 5.615 + 6.094}{6} \approx 5.07\%$
> >
> > •	CUB (10 steps)
> >
> > $\text{Avg. gain} = \frac{4.456 + 8.805 + 7.188 + 7.483 + 6.096 + 3.015}{6} \approx 6.17\%$
> >
> > For transparency, we also report the maximum gains:
> > KCA achieves 1.94% (ImageNet-R) and 7.52% (CUB), whereas DeL achieves 6.42% and 8.805%, respectively. These comparisons ensure that DeL is evaluated fairly using the same plug-in perspective as KCA.
> >
> > **MOS**
> >
> > To ensure a fair comparison, we retrain MOS with and without DeL under the same environment and VTAB datasets. Therefore, the reported numbers differ from those in the original MOS paper. All experimental protocols, datasets, and hyperparameters are now fully detailed in the revised supplementary file, with reproducible settings provided.
> >
> > ## Weakness (4): Figure 2 may be misleading
> > Thank you for your concern. Although Figure 2 serves as a conceptual illustration, **it is not an arbitrary or hypothetical schematic**. The visualized activations are directly extracted from real model runs, and the dendritic interactions are produced exactly by the computations formalized in our method. Thus, the figure reflects the actual modulation dynamics observed during inference, rather than an idealized depiction.
> >
> > Regarding the comparison with attention modules, the intent of Figure 2 is not to claim that any module with a similar parameter scale is inferior, but to highlight that DeL modulates signals locally at the dendrite level, instead of applying global token–token interaction like attention.
> >
> > ## Weakness (6): Stability implications
> > First, although DeL introduces extra parameters, the increase is small and scales linearly with the number of dendritic branches M. Our parameter analysis shows that as M increases, model performance remains stable and does not degrade, indicating that parameter growth does not introduce instability in practice.
> >
> > Second, we provide gradient computation details in the supplementary file. Because each dendritic weight is modulated by a sigmoid function, its gradient satisfies, $0 < \Delta w_{i,j} < \frac{1}{16}$.
> > Thus, regardless of the number of DeL branches, all gradients remain strictly bounded, ensuring numerical stability regardless of parameter size.
> >
> > Third, unlike standard backpropagation, each dendritic branch updates only its own synapses, without affecting others. Therefore, increasing the number of branches increases representational capacity but does not introduce cross-branch interference. This localized update rule further stabilizes learning both theoretically and empirically.
> >
> > In summary, DeL’s additional parameters do not compromise stability. Stability is intrinsically maintained through bounded gradients and localized branch-wise learning, validated mathematically and experimentally.

---

### Official Review · Reviewer_CS3z · 2025-10-31

**Soundness:** 3
**Presentation:** 3
**Contribution:** 3
**Rating:** 6
**Confidence:** 4

**Summary:**

Dendritic Learning (DEL) is a biologically inspired framework that mitigates catastrophic forgetting in class-incremental learning (CIL). It reduces homogeneous representations by introducing synaptic plasticity and multi-branch dendritic processing. The framework is model-agnostic, allowing integration with existing CIL methods by replacing their final fully connected layer with a dendritic block composed of four layers: synaptic, dendritic, membrane, and somatic. DEL extracts discriminative features, enhancing adaptability and retention across sequential learning tasks. The authors conducted experiments using a ResNet-18 backbone on standard CIL benchmarks demonstrating improved performance and reduced forgetting.

**Strengths:**

DEL framework is inspired by dendritic neuron models (DNMs) that mitigates catastrophic forgetting in CIL to balance stability and plasticity, fostering better generalisation across sequential tasks.

The authors provide comprehensive Grad-CAM visualisations that focus on relevant object regions across incremental tasks highlighting its capability for heterogeneous, discriminative, and class-specific feature learning.

DEL framwork replaces the fully connected layer with a four-layer dendritic block, where Synaptic layers handle excitatory and inhibitory inputs using learnable sigmoids for synaptic plasticity, Dendritic layers perform multiplicative (logical AND) operations, Membrane layers aggregate signals via weighted fusion (logical OR), and Somatic layers apply threshold-based classification.

The framework is model-agnostic, lightweight, and can be integrated into various CIL methods without using extra parameters, memory buffers, or rehearsal mechanisms.

**Weaknesses:**

While the proposed DEL framework mitigates catastrophic forgetting, several limitations reduce its overall impact. First, the method relies on fixed hyperparameters and a static architectural design that could hinder scalability to deeper backbones or multimodal tasks beyond the tested ResNet-18 and vision datasets.

DeL depends on backpropagation that undermines the claim of biological plausibility. This could introduce issues such as global error propagation, a limitation that has been seen in other neuron-inspired frameworks.

The fixed dendritic branch configuration (M = 2) may not generalise well across datasets, potentially constraining the balance between stability and plasticity.

From a theoretical standpoint, the paper lacks convergence proofs or a formal mathematical analysis of gradient behavior, particularly in high-dimensional spaces. The absence of such analysis limits confidence in the method’s robustness and stability.

Comparisons with more recent dendritic learning models are missing, which would help contextualize DeL’s novelty and contribution within the broader literature.

Although relative improvements are demonstrated, performance on complex datasets remains modest, suggesting that the approach is not yet mature

**Questions:**

How stable is DeL’s usage on large-scale datasets? Is there any trade-off between DeL’s biological plausibility and its ability to scale in modern deep networks such as Transfrmers. It has only been tested using ResNet18 backbone, have you done additional experiments on any other deeper versions of ResNET

Can DeL be extended to task-incremental, domain-incremental or online continual learning settings?

Given backpropagation's implausibility, have alternatives like evolutionary algorithms or localised updates been explored for training?

Why no direct comparisons to biologically plausible baselines like Dendritic Localized Learning (DLL) or dendritic SNNs? Do these yield similar gains?

---

> ### Author Response · Authors · 2025-11-19
>
> Dear reviewer, thank you very much for your detailed and constructive comments. We summarize and respond to each point individually as follows.
> ### 1.  Only ResNet18 was tested. Have you evaluated deeper backbones or more complex datasets?
> Our experiments actually span six datasets, including OmniBenchmark, whose complexity substantially exceeds typical small-scale benchmarks. Detailed descriptions are provided in the Supplementary File. To further broaden architectural diversity, we additionally evaluated ResNet34 and pretrained ViT. The results below show that DeL consistently improves performance across CNNs and large-scale pretrained transformers:
> | Backbone   | w/o DeL |   | w/Del |   |
> |------------|---------|---|-------|---|
> |            | $\bar{A}$| $A_L$ | $\bar{A}$| $A_L$ |
> | ResNet18   | 58.03 %  | 51.07 % | 63.85 % | 56.19 % |
> | ResNet34   | 55.26 %  | 47.79 % | 62.36 % | 51.32 % |
> |Pretrain ViT| 91.74 %  | 92.05 % | 92.10 % | 92.09 % |
> ### 2. DeL depends on backpropagation, which undermines biological plausibility
> We appreciate this insightful point. Indeed, BP is biologically implausible, and this limitation applies to all ANN-based models using gradient descent, including neuron-inspired frameworks. The biological plausibility of DeL arises not from the training rule, but from its architectural mechanisms.
> Neuroscientific studies show that dendrites exhibit localized nonlinear processing and spatially selective integration [1]. DeL incorporates these principles via:
>
> •branch-local nonlinearities in the dendritic layer,
>
> •position-dependent synaptic integration, and
>
> •localized plasticity reflecting dendritic compartmentalization.
>
> These mechanisms—not BP—provide DeL’s biological grounding. The resulting spatial selectivity enhances salient feature extraction and helps mitigate catastrophic forgetting, consistent with biological learning systems. A detailed biological explanation of the dendritic model is included in the Supplementary File.
> [1] Power-efficient neural network with artificial dendrite, *Nature Nanotechnology, 2020*
> ### 3. Fixed dendritic branch configuration (M=2) may not generalize
> We subsequently evaluated DeL across multiple models and consistently observed that $M = 2$ delivers the best overall performance, confirming its empirical robustness under diverse conditions.
> While a dynamically adjustable $M$ may offer additional flexibility for balancing stability and plasticity, implementing such a mechanism would significantly increase computational cost and undermine the high-efficiency design that makes DeL practical for continual learning. We agree that this is an interesting direction and plan to explore dynamic dendritic configurations in future work.
> ### 4. Lack of convergence proof or theoretical gradient analysis.
> Thank you for the comment. Full convergence guarantees in high-dimensional nonlinear systems are extremely challenging and rarely available for modern neural models. Instead, we provide robustness evidence through extensive empirical evaluation across dataset scales and model families, where DeL consistently yields stable improvements.
> In addition, we provide a formal mathematical analysis of the gradient behavior in the Supplementary File. For example, the update rule for the synaptic weights is given by:
> $\Delta w_{i,j}= \delta'(O)\cdot g_i\cdot F(I)(1-F(I))\cdot I\cdot S_{i,j}(1-S_{i,j})x_i$
> which further supports the theoretical soundness of DeL’s mechanism.
> ### 5. Can DeL be extended to task-incremental, domain-incremental or online continual learning settings?
> Yes. DeL is fully model-agnostic and can, in principle, be integrated into task-IL, domain-IL, or online CL settings. These extensions are theoretically feasible, and we plan to validate them empirically in future work.
> ### 6. Have you explored biologically plausible alternatives to BP (e.g., evolutionary algorithms or localized updates)?
> Thank you for the question. While ECs can outperform BP on small models, they remain computationally challenging for high-dimensional continual-learning settings. Scaling these methods is an important direction we plan to explore in future work.
> ### 7. Why no comparisons to biologically plausible baselines (e.g., DLL, dendritic SNNs)?
> Thank you for this valuable question. DLL and dendritic SNNs pursue biologically faithful learning rules, whereas DeL focuses on a different design goal: a computationally efficient, model-agnostic dendritic module that can be seamlessly integrated into modern CL pipelines. The two classes of methods address different dimensions of biological inspiration, so DLL/SNNs are not directly comparable baselines for our scope.
> To strengthen classifier-level comparisons among model-agnostic approaches, we additionally compared DeL with KAC. Across datasets and architectures, DeL consistently achieves larger gains.
> |           | KAC| DeL   |
> |------------|----------|---------|
> | ImageNet-R |  0.91% |  3.489% |
> | CUB200     |  5.90% |  7.391% |

---

> > ### Comment · Reviewer_CS3z · 2025-11-25
> >
> > Thank you for the detailed response and the new experiments. They've helped clarify some aspects of the work, however there are few still remaining:
> >
> > • My main concern centers on the experimental design used to compare the different backbones. While I appreciate the additional experiments, the core methodological issue remains. Comparing ResNets trained from scratch against a pretrained ViT is not a like-for-like comparison and introduces a critical issue regarding the training paradigm as this makes it impossible to disentangle the effect of pretraining from the architectural properties of CNNs versus Transformers. The counter-intuitive result of ResNet-18 outperforming ResNet-34 highlights that the from-scratch training dynamic is a dominant factor. To rigorously validate the model agnostic claim across different architectural classes, a controlled experiment is required, such as comparing a pretrained ResNet against the pretrained ViT. Without this, the central claim remains unsubstantiated under fair conditions.
> >
> > • I'm particularly interested in the interaction between DeL and the self-attention mechanism. Since attention heads are already powerful tools for creating diverse feature representations, what is the complementary benefit of adding the DeL block?
> >
> > • Regarding the static branch configuration (M=2), which you frame as a trade-off for efficiency, have you considered any low-cost heuristics to make it dynamic?
> >
> > • You mention that DeL is theoretically applicable to online CIL. What do you see as the primary challenge in a true online setting where data arrives one sample at a time? Would the synaptic plasticity mechanism in DeL be robust enough to learn stably from such noisy, single-instance updates?

---

### Official Review · Reviewer_7Yo3 · 2025-11-01

**Soundness:** 3
**Presentation:** 4
**Contribution:** 3
**Rating:** 6
**Confidence:** 3

**Summary:**

This paper introduces a novel neurobiology-inspired dendritic learning (DeL) module to address the challenges of catastrophic forgetting and homogeneous representations in class-incremental learning. The proposed DeL architecture, which consists of a Synaptic Layer, Dendritic Layer, Membrane Layer, and Somatic Layer, emulates synaptic plasticity through learnable weights to modulate feature strength. A key strength is its model-agnostic design, allowing it to be seamlessly integrated into vasrious mainstream CIL methods by replacing the standard classifier. Extensive experiments across six benchmark datasets demonstrate that DeL consistently improves the average incremental accuracy of baseline models. Furthermore, the authors provide a thorough analysis using Grad-CAM visualizations, weight distribution analysis, information entropy, and class selectivity, which improve that DeL mitigates forgetting by reducing parameter drift and increasing the proportion of class-specific neurons.

**Strengths:**

This paper analyzes the multi-branched dendrites and synaptic plasticity of biological neurons, transforming them into a clearly structured, integrable artificial neural network module. The approach is relatively novel. Besides，This paper benchmarks DeL against a wide range of CIL methodologies on multiple
datasets, providing strong empirical support for its effectiveness. In addition, the detailed visualizations significantly enhance the paper’s persuasiveness and interpretability.

**Weaknesses:**

The paper identifies homogeneous representations as a limitation of existing methods and compares DeL with several CIL paradigms. However, it lacks detailed discussion and comparison with other CIL methods that also explore representation heterogeneity or biologically inspired plasticity. Including such comparisons could further highlight the unique advantages of DeL.Although the paper claims that DeL reduces memory usage and relies on fewer discriminative features, it lacks a systematic analysis of the computational and parameter overhead introduced by the DeL module itself (such as increases in parameter count, FLOPs, or training/inference time) compared with baselines. Adding this analysis would strengthen the paper’s credibility.Currently, the experiments are limited to relatively small-scale datasets. Evaluations on large-scale datasets (e.g., ImageNet-1K) and broader architectures (e.g., ViT) would make the empirical validation more convincing.

**Questions:**

The ablation studies mainly focus on synaptic plasticity (Table 2). Could the authors provide additional ablations for other key components of DeL (such as the weighted fusion mechanism in the Membrane Layer and the threshold adjustment in the Somatic Layer) to more comprehensively demonstrate each component’s contribution?

The paper claims that DeL is particularly valuable in non-rehearsal settings where rehearsal buffers are limited. Are there experiments directly comparing DeL’s performance under “with-buffer” and “without-buffer” conditions to quantitatively support this claim?

In Table 1 and Figure 3, the dataset names (CIFAR100 vs. CIFAR B0 Inc10) appear inconsistent. It is recommended to unify the dataset naming to improve clarity and reduce potential confusion.

---

> ### Author Response · Authors · 2025-11-19
>
> Dear reviewer, thank you very much for your detailed and valuable comments. We summarize and respond to each of your concerns below.
> ### 1. Ablations of membrane and somatic layers
> Ablations on all components (synaptic, dendritic, membrane, somatic) were conducted. Table 2 shows representative results; full experiments are in the Supplementary File. Replacing membrane fusion or somatic thresholding with simple activations (e.g., sigmoid/tanh) collapses dendritic diversity and significantly degrades performance, confirming the necessity of these layers.
> ### 2. Problem Framing (rehearsal vs. non-rehearsal)
> We used the non-rehearsal setting only to define class-IL. Our method is not limited to non-rehearsal setups. Experiments include both settings; full configs are in the Supplementary File.
> ### 3. Dataset naming inconsistency
> Thank you for pointing this out. We have corrected all naming inconsistencies in the revised manuscript.
> ### 4. Limited dataset scale and architectural diversity
> Our experiments cover six datasets, including OmniBenchmark, which is considerably more challenging than typical small-scale benchmarks. Detailed dataset descriptions are provided in the Supplementary File.
> To address architectural diversity, we additionally evaluated pretrained ViT and ResNet34. DeL consistently improves performance across these architectures, including large transformer-based models:
> | Backbone   | w/o DeL |   | w/Del |   |
> |------------|---------|---|-------|---|
> |            | $\bar{A}$| $A_L$ | $\bar{A}$| $A_L$ |
> | ResNet18   | 58.03 %  | 51.07 % | 63.85 % | 56.19 % |
> | ResNet34   | 55.26 %  | 47.79 % | 62.36 % | 51.32 % |
> |Pretrain ViT| 91.74 %  | 92.05 % | 92.10 % | 92.09 % |
> ### 5. Computational and parameter overhead
> We reported computational complexity in the Supplementary File. For clarity, the key comparison is shown below: DeL adds only 0.4% parameters but yields up to 9% gains.
>
> | Methods |  CUB200  | VTAB    |  CUB200  | VTAB    |
> |---------|----------|---------|----------|---------|
> |         | Params   | Params  |  ACC     |   ACC  |
> | iCaRL   |  10.75 M | 10.68 M |  31.01 % | 58.09 % |
> | w/ Del  |  10.84 M | 10.70 M |  40.33 % | 62.49 % |
> |TagFex   |  26.82 M | 25.96 M |  51.08 % | 61.79 % |
> |w/ Del   |  27.82 M | 26.09 M |  55.43 % | 64.41 % |

---

> > ### Comment · Reviewer_7Yo3 · 2025-11-23
> >
> > Thank you very much for the authors’ response, which resolves most of my concerns. However, several issues still remain:
> >
> > 1. The authors still have not evaluated the method on larger-scale or longer-horizon data.
> > As we all know, continual learning fundamentally requires verifying whether the model’s performance remains stable under long-term sequential updates. This concern has not yet been addressed.
> >
> > 2. The authors claim that DEL reduces memory usage, but the provided evidence does not directly support this claim.
> > Specifically, the authors show that DEL adds only 0.4% parameters while improving performance by up to 9%. While this demonstrates the advantage of the method, it does not directly demonstrate reduced memory consumption. Therefore, my original concern remains unresolved.
> >
> > 3. Regarding computational overhead:
> > I would very much like to see a comparison of DEL’s FLOPs or training/inference time relative to other baselines. Without this information, it is difficult to assess the practical efficiency of the method.

---

> > > ### Author Response · Authors · 2025-11-25
> > >
> > > ## The authors still have not evaluated the method on larger-scale or longer-horizon data.
> > >
> > > We fully agree that continual learning should be evaluated under extended sequential scenarios. To address this concern, we conducted longer-horizon experiments ranging from 5 to 40 sequential tasks, as summarized in Table 7 of the supplementary file. The results show that DeL consistently improves performance across all baselines as the learning horizon increases, indicating stable long-term learning behavior. For example, DER achieves an average accuracy of 30.87 after 40 steps, whereas DER + DeL maintains a significantly higher accuracy of 34.76, clearly mitigating cumulative degradation during prolonged learning. The average accuracies $\bar{A}$ across different horizons are shown below,
> > >
> > > |          | 5   step | 10 step | 20 step | 40 step |
> > > |----------|:--------:|:-------:|:-------:|:-------:|
> > > | Finetune |   28.57  |  19.48  |  11.18  |   7.21  |
> > > | 　w/ DeL |   **32.80**  |  **23.68**  |  **14.67**  |   **8.28**  |
> > > |  Replay  |   41.31  |  30.59  |  23.03  |  20.59  |
> > > | 　w/ DeL |   **46.65**  |  **39.25**  |  **32.44**  |  **24.00**  |
> > > |    LwF   |   32.77  |  22.75  |  14.74  |   9.31  |
> > > | w/ DeL   |   **36.34**  |  **29.73**  | **18.73**  |  **11.10**  |
> > > |   iCarL  |   39.95  |  31.73  |  22.59  |  21.00  |
> > > | 　w/ DeL |   **45.77**  |  **36.30**  |  **31.21**  |  **26.20**  |
> > > |    DER   |   48.61  |  39.29  |  34.29  |  30.87  |
> > > | 　w/ DeL |   **54.20**  |  **43.62**  |  **40.67**  |  **34.76**  |
> > >
> > > ## The authors claim that DEL reduces memory usage, but the provided evidence does not directly support this claim.
> > >
> > > We believe there is a misunderstanding. Our intention in proposing DeL is to enable models to extract fewer discriminative features for each category, thereby reducing the model’s **feature-level memory requirements**, rather than decreasing **hardware memory usage**.
> > >
> > > Real-world problems are typically linearly inseparable, and accurate classification in high-dimensional feature spaces often requires strong nonlinear capability. If a model can achieve precise classification using fewer, non-redundant discriminative features, it inherently reduces the amount of feature information that must be stored to support correct classification, thereby lowering memory demands at the representational level.
> > >
> > > Assuming that key features across different categories are complementary, a model that can retain fewer essential features while maintaining the same feature dimension will more easily separate categories and thus facilitate continual learning. Owing to DeL’s strong nonlinear capacity, it effectively reduces the number of key features that must be memorized for each category. This reduction in feature dependence subsequently alleviates catastrophic forgetting.
> > >
> > > ## Regarding computational overhead.
> > > We have provided a detailed comparison of model parameters and training time in the supplementary material. Please refer to the supplementary file. As shown in Tables 10 and 11, incorporating DeL introduces only a marginal increase in runtime across all datasets and methods, while consistently improving performance. This demonstrates that the computational overhead of DeL is minimal relative to its performance gains.

---

### Official Review · Reviewer_M6Ph · 2025-11-02

**Soundness:** 2
**Presentation:** 3
**Contribution:** 2
**Rating:** 2
**Confidence:** 4

**Summary:**

The authors proposes a biologically inspired classifier head for class-incremental learning called DeL. DeL replaces the standard fully connected layer with a four-stage module (synaptic gating, dendritic branching, membrane fusion, and somatic activation) intended to increase feature diversity and reduce forgetting. The module is evaluated as a plug-in to several existing CIL methods on small to medium-scale benchmarks, showing some rather small accuracy gains. While the implementation is general, the approach offers limited conceptual novelty: it functions as a gated multi-branch MLP trained with standard backpropagation, and its connection to biological learning is superficial. The experimental analysis lacks statistical rigor (single seed, no forgetting metrics, limited architectures) and omits comparisons to strong modern baselines such as cosine, prototype, or prompt-based heads. Overall, the work seems rather incremental and it does not meet the novelty and depth expected for ICLR acceptance.

**Strengths:**

a. Simple, modular design that can replace a standard classifier layer in many CL frameworks
b. Broad empirical evaluation across several CIL methods and datasets showing generally consistent (but small) improvements.
c. Clear architectural description with ablation studies isolating some effects
d. Use of visual analyses (Grad-CAM, weight distributions) to illustrate feature diversity and reduced forgetting

**Weaknesses:**

1. The main weakness, at least in my opinion is in terms of novelty. DeL is effectively a gated multi-branch MLP head with standard backpropagation. This is nothing new fundamentally.
2. The “biological plausibility” claim is rather superficial. There is no local learning or biologically grounded mechanism
3. The experimental section lacks rigor. If I understand well the authors use a single seed, no statistical variation, no forgetting or BWT metrics. Plus, only small datasets
4. The problem framing is confusing/nconsistent: the paper claims a non-rehearsal setup but evaluates with rehearsal-based methods
5. Missing comparisons to strong and directly relevant baselines (cosine, prototype, MoE, prompt-based, or attention-based heads).
6. Ambiguity in classifier output. Unclear whether sigmoid or softmax is used for multi-class loss. The training objective not well defined.
7. The design of “membrane” and “somatic” layers is rather unclear. No ablation or justification for these components.
8. Limited architectural scope: only ResNet-18 tested. Unclear generalization to pretrained or transformer backbones.
9. Qualitative visualizations are rather unconvincing and they do not substantiate claims of reduced forgetting or improved representation diversity.

**Questions:**

First, I suggest that the authors also read the following papers because they are quite relevant -- even though they may not (all) be neuro-inspired:

Learning to Prompt for Continual Learning
DualPrompt: Complementary Prompting for Rehearsal-Free Continual Learning
Mixture-of-Experts Meets Prompt-Based Continual Learning
Learning a Unified Classifier Incrementally via Rebalancing
Few-Shot Class-Incremental Learning via Training-Free Prototype Calibration
Dynamic Adapter Tuning for Long-Tailed Class-Incremental Learning
A rapid and efficient learning rule for biological neural circuits
Orthogonal Subspace Learning for Continual Adaptation

Some additional comments and suggestions:

- plz clearly define the loss function and output layer semantics. If sigmoid activations are used, explain how multi-class probabilities are computed and justify the choice over softmax
- plz report results over multiple random seeds and include standard metrics for continual learning (average accuracy, forgetting, backward transfer). This is all standard practice.
- quantify the computational and parameter overhead of DeL compared to a linear classifier.
- I suggest you add ablations for the membrane and somatic layers to show their individual contribution
- Include comparisons to modern classifier-head designs such as cosine-normalized, prototype-based, attention pooling, and small MLP or gating heads with equal parameter count
- Important to test DeL on stronger and more diverse architectures, especially pretrained ViTs, to assess generality.
- Plz either rephrase or substantiate the “biologically plausible” claim. For example you can do so by introducing a biologically inspired local learning rule or by avoiding that terminology altogether
- Improve the clarity of experimental settings: specify buffer sizes, rehearsal policies, and dataset protocols
- Perform a rehearsal-free evaluation to verify whether DeL alone improves plasticity–stability balance.

---

> ### Author Response · Authors · 2025-11-19
>
> Dear reviewer, thank you very much for your detailed comments. Below we summarize and respond to each point.
> ### 1. Novelty Concerns: “DeL is just a gated multi-branch MLP.”
> We respectfully disagree. Extensive neuroscience shows that dendrites are nonlinear computational units that perform local computations, generate branch-specific spikes, and act as independent subunits [1–3]. Modern neuromorphic/ANN research further shows that dendritic morphology and localized nonlinear integration improve discriminability, robustness, and efficiency [4–7].
> DeL is directly inspired by these principles: it introduces branch-local nonlinearities, position-dependent synaptic integration, and localized plasticity, rather than global mixing as in MLPs. It is also model-agnostic and easily pluggable into CL pipelines, a capability not available in prior dendritic models. The novelty lies in translating biologically grounded dendritic mechanisms into a practical CL module, which, to our knowledge, has not been explored. We have added a concise dendritic-learning review in the manuscript.
>
> References: [1] Nature 1997; [2] Nat. Neurosci. 2015; [3] Science 2020; [4] Nat. Electron. 2024; [5] Nat. Commun. 2025; [6] NeurIPS 2018; [7] Nat. Commun. 2023; [8] Nat. Rev. Neurosci. 2020.
> ### 2. Biological plausibility is superficial
> We acknowledge that BP is biologically implausible (as in all gradient-based methods). However, DeL’s plausibility comes from its architecture, not its training rule. DeL incorporates dendritic local nonlinearity and position-dependent integration [1], enabling localized plasticity and selective feature representation, which empirically reduces forgetting. We also added a detailed dendritic-model explanation in the Supplementary File.
> ### 3. Experimental Rigor: (1) Seeds; (2) Computation
> (1) We added experiments with seeds {1993, 1994, 1995}. DeL consistently improves performance across datasets; updated averages are shown in Table 1 and full results in the Supplementary File.
> (2) We reported computational complexity in the Supplementary File. For clarity, the key comparison is shown below: DeL adds only 0.4% parameters but yields up to 9% gains.
> | Methods |  CUB200  | VTAB    |  CUB200  | VTAB    |
> |---------|----------|---------|----------|---------|
> |         | Params   | Params  |  ACC     |   ACC  |
> | iCaRL   |  10.75 M | 10.68 M |  31.01 % | 58.09 % |
> | w/ Del  |  10.84 M | 10.70 M |  40.33 % | 62.49 % |
> ### 4. Problem Framing (rehearsal vs. non-rehearsal)
> We used the non-rehearsal setting only to define class-IL. Our method is not limited to non-rehearsal setups. Experiments include both settings; full configs are in the Supplementary File. DeL also improves rehearsal-free methods such as LwF:
> | Method | CIFAR100         | CUB200           | ImageNet-R       |  Omni             |
> |--------|------------------|------------------|------------------|------------------|
> | LwF    | 46.56±0.56       | 24.34±2.24       | 20.24±1.18       |  38.40±0.55       |
> | w/ DeL | **52.98±0.79**   | **31.53±1.47**   | **25.71±1.73**   |  **41.94±0.73**   |
> ### 5. Missing Baselines; Limited Architectures
> We compare against strong baselines. TagFex already includes attention-based heads; replacing them with DeL consistently improves results. DeL also outperforms MLP-head models such as LwF and DER.
> To broaden architecture coverage, we added pretrained ViT and ResNet34 experiments:
> | Backbone   | w/o DeL |   | w/Del |   |
> |------------|---------|---|-------|---|
> |            | $\bar{A}$| $A_L$ | $\bar{A}$| $A_L$ |
> | ResNet18   | 58.03 %  | 51.07 % | 63.85 % | 56.19 % |
> | ResNet34   | 55.26 %  | 47.79 % | 62.36 % | 51.32 % |
> |Pretrain ViT| 91.74 %  | 92.05 % | 92.10 % | 92.09 % |
> ### 6. Classifier Output Semantics (sigmoid vs. softmax)
> The somatic layer applies independent sigmoids to each class-specific output, resulting in $N$ logits for $N$ classes. Based on these logits, we use cross-entropy for multi-class classification. Detailed formulas are provided in the Supplementary File. All baselines follow their original loss settings for fair comparisons.
> ###  7. Membrane & Somatic Layer Ablation Study
> Ablations on all components (synaptic, dendritic, membrane, somatic) were conducted. Table 2 shows representative results; full experiments are in the Supplementary File. Replacing membrane fusion or somatic thresholding with simple activations (e.g., sigmoid/tanh) collapses dendritic diversity and significantly degrades performance, confirming the necessity of these layers.
> ### 8. Qualitative Visualizations Not Convincing
> Following [9], we analyze information entropy, class selectivity, and weight distributions. DeL reduces mixed selectivity and enhances class-specific tuning, providing quantitative evidence for improved representation robustness and reduced forgetting.
>
> [9]  *Nat. Commun.*, 2025.

---

> > ### Comment · Reviewer_M6Ph · 2025-11-23
> >
> > I appreciate the authors' responses and the additional results but I still believe that, if we remove from the paper the connections with neuroscience (which are also superficial) and evaluate it strictly as an ML contribution, it is not something really new. I have already given relevant references (even though it is not an exhaustive list) in my main review.

---

> ### Author Response · Authors · 2025-11-26
> **Official Comment by Authors -- Clarification (1)**
>
> We sincerely appreciate the reviewer’s further comments. The reviewer questioned the biological relevance of DeL, suggesting that if the neuroscience connections were ignored, the machine learning contribution alone may not be sufficiently novel. We respectfully offer the following **4 clarifications**:
>
> ## 1.  On the claimed “ superficial” relevance to neuroscience
> We do not consider the connection between DeL and neuroscience to be superficial. The core mechanism of DeL explicitly leverages dendritic nonlinear integration and branch-specific learning to enhance representations and improve resistance to forgetting.
>
> To illustrate why dendritic nonlinearities provide expressive advantages beyond classical MLP summation, consider two inputs $A$ and $B$ with values $x_A = 1, x_B = 2$. Standard MLP aggregation computes
> $y = \sum_{i} w_i x_i$ .
> If the MLP has two sub-branches, both homogeneous aggregation, $y = (2+2) + (1+1) = 6$, and mixed aggregation, $y = (1+2) + (1+2) = 6$,
> produce identical outputs regardless of whether similar features are clustered or mixed. This results in homogenized representations that hinder discriminability.
>
> In contrast, if dendritic sub-branches apply local nonlinear transformations,
> $y_d = \sum_k f(z_k), \quad \text{with } f(z) = z^2$,
> then the same inputs yield different outputs depending on how they are allocated to dendritic branches:$y_d = (2+2)^2 + (1+1)^2 = 20,$ whereas for mixed inputs, $y_d = (1+2)^2 + (1+2)^2 = 18$.
> Thus, **synaptic location becomes part of the memory**, enabling DeL to selectively encode feature clusters and **reduce representational homogenization**. This position-sensitive nonlinear integration is a biologically inspired computational mechanism, rather than a mere engineering trick.
>
> Returning to the essence of continual learning, real-world problems are typically **nonlinearly inseparable**. Models therefore require strong nonlinear capabilities to accurately distinguish classes in high-dimensional feature spaces. During continual learning, model parameters drift when learning to new tasks, weakening nonlinear separability for earlier tasks and causing catastrophic forgetting. If a model can achieve accurate classification using **fewer non-redundant discriminative features**, it reduces the number of features that must be preserved, thereby lowering memory demands. Assuming discriminative features across categories are complementary, a model that requires fewer such features, while maintaining similar representational capacity, can more easily discriminate among more categories over time, enabling continual learning.
>
> Based on this principle, DeL leverages **dendritic nonlinearities to become sensitive to synaptic input locations**, allowing the model to **extract fewer discriminative features** for a given class. This reduces representational memory requirements and effectively mitigates catastrophic forgetting.

---

> ### Author Response · Authors · 2025-11-26
> **Official Comment by Authors -- Clarification (2)**
>
> ## 2. Comparison with prior continual learning literature
>
> As indicated by the references you provided, most existing approaches propose various strategies to mitigate parameter drift and alleviate catastrophic forgetting in continual learning. L2P [1] incorporates a prompt pool to mitigate forgetting under a non-rehearsal setting, where additional instance-wise prompts serve as exemplars of old knowledge. DualPrompt [2] introduces two complementary prompt spaces to further improve performance. NoRGa [3] abandons a shared pool and instead adopts a Mixture-of-Experts (MoE) architecture, enhancing scalability through expert partitioning. The method in [4] tackles class-incremental learning by jointly applying cosine normalization, less-forgetting feature constraints, and inter-class margin rebalancing, thereby mitigating decision boundary bias and catastrophic forgetting without rehearsal. Similarly, [5] addresses few-shot class-incremental learning without training by calibrating class prototypes through semantic-aware similarity, confidence-driven smoothing, and balanced prototype aggregation, which alleviates few-shot bias and boundary shift without feature rehearsal. DAT [6] adopts dual distillation across both tasks and classes to preserve previously learned knowledge, while other methods compute the semantic similarity between new and old class prototypes and perform weighted fusion to address biased representations of novel categories and the imbalance between new and old data.
>
> Although effective, these models do not fundamentally solve continual learning at **the architectural level**; instead, they rely on external regularization or distillation to “suppress forgetting.” In contrast, DeepMind’s recent work [7] explores the potential of biological neural systems and proposes a biologically gated network, where dendritic gating and local learning replace traditional backpropagation, forming a more biologically plausible learning mechanism.
>
> In comparison, DeL takes a further step by emphasizing how dendritic structures themselves contribute to learning and representation. Specifically, DeL enhances **dendritic nonlinearity and branch-wise independent learning**, thereby improving representational capacity and resistance to forgetting. More concretely:
>
> •	Nonlinear dendritic branches selectively store category-specific features, reducing cross-class interference;
>
> •	Independent branch learning enables different categories to form separated representations within distinct dendritic substructures;
>
> •	Synaptic plasticity allows DeL to dynamically regulate dendritic connections.
>
> As explained in Section B of the supplementary file, synaptic plasticity in DeL exhibit four states: constant-zero, constant-one, inhibitory, and excitatory. Among them, **constant-zero and constant-one enable dynamic integration of dendritic branches**, providing powerful nonlinear expressiveness. This dual mechanism of structure and learning strongly demonstrates the tight connection between DeL and neuroscience principles. Benefiting from this strong nonlinear capacity, DeL can use fewer key features to achieve accurate classification, reducing representational competition and thereby substantially mitigating catastrophic forgetting.
>
> Additionally, we integrated DeL into L2P [1] and DualPrompt [2] for performance evaluation, yielding further enhanced results. **This demonstrates that our research does not compete with these approaches but instead complements them, highlighting both the necessity and the immense potential of DeL-based research.**
>
> |            | CIFAR100 |       |       | CUB200 |        |       |
> |------------|----------|-------|-------|--------|--------|-------|
> |            | $\bar{A}$        | BWT   | FWT   | $\bar{A}$ | BWT    | FWT   |
> | L2P        | 88.3     | -6.48 | -1.18 | 74.86  | -9.98  | -1.46 |
> | w/ DeL     | 90.19    | -5.01 | -1.78 | 79.62  | -6.48  | -1.58 |
> | DualPrompt | 86.95    | -6.62 | -2.93 | 79.38  | -12.78 | -1.26 |
> | w /DeL     | 89.36    | -4.19 | -1.72 | 84.55  | -9.50  | -1.49 |
>
>
> [1] Learning to Prompt for Continual Learning, *CVPR, 2022*
>
> [2] DualPrompt: Complementary Prompting for Rehearsal-Free Continual Learning, *ECCV, 2022*
>
> [3] Mixture-of-Experts Meets Prompt-Based Continual Learning, *NeurIPS, 2024*
>
> [4] Learning a Unified Classifier Incrementally via Rebalancing, *CVPR, 2019*
>
> [5] Few-Shot Class-Incremental Learning via Training-Free Prototype Calibration, *NeurIPS, 2023*
>
> [6] Dynamic Adapter Tuning for Long-Tailed Class-Incremental Learning, *WACV, 2025*
>
> [7] A rapid and efficient learning rule for biological neural circuits, *BioRxiv, 2021*

---

> ### Author Response · Authors · 2025-11-26
> **Official Comment by Authors -- Clarifications (3)&(4)**
>
> ## 3. Supporting evidence from Harvard University
> Furthermore, a recent study [1] from **Harvard University** has mathematically demonstrated, for the first time, that **dendritic structure** can alter learning rules rather than merely increase expressive capacity. This work elucidates how dendritic nonlinearities influence synaptic learning dynamics. The authors proved that:
>
> •	With the same number of synapses, nonlinear dendritic branches enable faster and more robust learning;
>
> •	Dendritic nonlinearity optimizes synaptic resource allocation;
>
> •	Local synaptic clusters naturally form “feature groupings.”
>
> Notably, the third finding—that local synaptic clusters naturally form feature groupings—**aligns well with our analysis in Section B of the supplementary file.** Specifically, DeL enables local synaptic clusters to assist model discrimination through distinct synaptic states, providing stronger evidence for the functional importance of dendritic structure in DeL.
>
> [1] Impact of Dendritic Nonlinearities on the Computational Capabilities of Neurons, *PRX Life, 2025*
>
> ## 4. On the importance of interdisciplinary innovation
> We agree that evaluating machine learning contributions independently is reasonable. Research should maintain diversity. Yet, we believe interdisciplinary advances are essential to avoid stagnation. Influential ML components such as momentum-based optimizers (physics) and spiking networks (biology) were initially considered niche inspirations but later became impactful computational tools. Similarly, dendritic computing has increasing evidence of providing fundamental computational benefits. DeL is a step in this direction: a biologically grounded architectural unit aimed at improving continual learning.

---

> > ### Comment · Reviewer_M6Ph · 2025-11-26
> > **(Responding to all four previous points)**
> >
> > I appreciate the detailed additional argumentation provided by the authors. However, the issues raised in my review remain unaddressed IMO from an ML perspective. The paper’s  motivation draws heavily from neuroscience but the proposed module is still functionally close to existing gated or multi-branch classifier heads trained with standard backpropagation. The paper does not provide clear demonstration of conceptual novelty relative to well established CIL architectures.

---

### Note · Authors · 2026-02-04

I have read and agree with the venue's withdrawal policy on behalf of myself and my co-authors.

---

### Meta-Review · Area_Chair_y7ZN · 2026-01-06

**Summary:**

This paper proposes DeL, a biologically inspired dendritic-style classifier head for class-incremental learning. It replaces the standard linear classification head with a bio-inspired module containing synaptic layer, dendritic layer, membrane layer, and somatic layer.

The reviewers are commonly concerned by whether the proposed method provides sufficient novelty and technically rigorous contribution, and whether the experimental evidence is solid and fairly controlled. Reviewer M6Ph argues that the proposed module is functionally close to gated multi-branch head, and the biological plausibility claim is superficial. This makes the novelty limited and the claims questionable. Reviewer M6Ph, CS3z, and Utud mentioned the connection with other related methods, including standard attention layer and some classifier design (in reviews of Utud),  and requested experiments, which also reflects the limited and unclear novelty.

The multiple reviewers also raise significant concerns about the insufficient evaluations and baselines and non-rigorious experimental design. The specific concerns include insufficient ablations and comparison with baselines (M6Ph,CS3z,Utud), absent comparison with related works (M6Ph), insufficient datasets and backbones (7Yo3), lack of comparisons with related works, e.g., other bio-inspired works and related CIL methods (M6Ph, CS3z, and Utud). The insufficient scope and rigor of the evaluation further weaken the justification of the claimed advantages.

Regarding the proposed bio-inspired module and the claim of superiority over standard linear classifier heads, the AC remains concerned. The authors argue that standard linear layers contribute to catastrophic forgetting and that the proposed bio-inspired head functionally mitigates forgetting. However, in the experiments, the model backbone (e.g., ResNet-18) is still trained end-to-end with standard convolutional layers and backpropagation, with only the final classifier replaced. This makes it unclear to what extent the reported gains can be attributed to the proposed “bio-inspired” mechanism rather than to generic increases in head capacity or architectural modifications, and it weakens the stated causal link between linear heads and forgetting.


Additional concerns regarding unclear details were also raised, which are largely connected to the main issues discussed above. While the authors provided further discussion and additional experiments to address these points, the core concerns related to limited novelty, biological plausibility, and the adequacy and fairness of the evaluation remain unresolved. Accordingly, the AC recommends rejection.

**Reviewer Concerns:**

Concerns related to specific technical details are addressed or partially addressed through the rebuttal, including additional experiments with multiple random seeds, clarification of the loss function and output-layer semantics, and reporting of computational complexity.

However, as discussed more in detail above, the primary concerns regarding fundamental conceptual novelty, the rigor of the design, and the adequacy and fairness of the evaluation remain unresolved. While further discussion and some additional experiments are provided, these do not sufficiently resolve the core issues, which therefore persist.

**Reviewer Scores:**

- Reviewer M6Ph. Explicitly states multiple times that novelty concerns and the related concerns for the claims and experiments. Despite added experiments, the concerns remain. The reviewer would likely maintain the original rejection-2 score.

- Reviewer 7Yo3. Indicates most concerns resolved; remaining issues (long-horizon, memory/overhead interpretation) were not fully addressed with added long-horizon and runtime discussion. The score (6) tends to be maintained with low probability to be improved.


- Reviewer CS3z. Appreciates additional experiments but raises a substantial fairness/control issue about pretrained vs scratch comparisons and wants more controlled validation; likely remains 6 but with reduced confidence or reduced score.


- Reviewer Utud. Raises strong concerns about traceability of rebuttal results, missing clarity, and insufficient theoretical support; even after author rebuttal, these concerns are serious and likely keep the score at reject-2.

---

### Decision · Program_Chairs · 2026-01-26

Reject